# Causal Model of Participation, Perceived Enjoyment, and Learning Attitudes in “the 0th Period Physical Education Class” of Middle Schools in South Korea

**DOI:** 10.3390/ijerph18147668

**Published:** 2021-07-19

**Authors:** Wonjae Jeon, Chanwoo Ahn, Heonsu Gwon

**Affiliations:** Department of Senior Sports Course, Daegu Haany University, 1 Haanydaero, Gyeongsan-si 38610, Korea; dreamj007@hanmail.net

**Keywords:** 0th period physical education class, perceived participation, learning attitude

## Abstract

This study aims to establish the basis for the institutional implementation of the 0th period physical education class to promote the health and academic performance of Korean teenagers. To achieve this goal, this study determined the impact of middle school students’ participation in physical activities during the 0th period on perceived enjoyment and learning attitude. To examine the model, 282 questionnaires were collected from middle school students in a metropolitan city in South Korea. The samples were obtained using the convenience sampling method, and correlation analysis and structural equation modeling were performed using SPSS 21.0 and Amos 21.0. The findings are as follows: first, the participation of middle school students in physical activities during the 0th period had a statistically significant effect on perceived enjoyment. Second, perceived enjoyment had no statistically significant effect on learning attitude. Third, participation was shown to have a significant effect on learning attitudes. These findings supported the academic basis for the implementation of the 0th period physical education class for middle school students and application of practical measures to encourage their participation.

## 1. Introduction

In Korean society, physical education has not attracted much attention in regard to school education. However, recently, the importance of physical education in schools has increased. This is because the value of school sports is being highlighted to prevent various social problems arising in the Korean education system, such as excessive competition for entrance exams, school violence, and suicide [1]. On 20 December, 2011, a 13-year-old middle school student committed suicide in an apartment in Daegu, South Korea, as he was being bullied by two other students. This incident sent the whole nation into a state of shock [2].

Kingdon [3] suggested that, in a critical events or crisis, a major social problem can be emphasized and formed into a public agenda. Accordingly, in 2012, the Ministry of Education announced “seven major practical tasks for eradicating school violence” and implemented “middle school sports club activities” [4]. For the past decade, the policy has contributed to the physical, mental, and social development of students by encouraging them to voluntarily participate in activities, and has been positively evaluated as fostering smooth interpersonal relationships and positive self-identity [5,6,7].

However, as the educational and social atmosphere of Korean society remains centered on high school and college entrance exams, it is highly likely that students will not only be under intense pressure from childhood to adolescence, but will also be exposed to stress, threatening their physical and mental health [8,9]. Furthermore, Korean students spend most of their time at school, so there is an urgent need to encourage physical activity at school. Nevertheless, the number of playgrounds in Korean schools are decreasing or disappearing [10]. Moreover, decreasing physical activity, the obesity rate, and smartphone addiction are evident among teenagers [1,8].

As mentioned above, most parents do not prioritize their children’s physical activities over their studies because Korean society is very passionate about education. This is becoming one of the major factors hindering the expansion of physical activities in adolescents. Nevertheless, with the introduction of the “0th period physical education class” through mass media, the academic effects of early morning exercise was shared in the Korean sports community. In 2005, it began at Naperville Central High School, Illinois, US. This class scientifically proved that, before starting regular classes in the morning, physical activity contributed to brain balance and functioning, improved concentration, memory, and attitude in regular classes [11]. Moreover, academic interest in the school has peaked since the media reported on the school conducting the “0th period physical education class” through MBC’s “Brain Revolution Project” in 2010 [12].

As such, it is necessary to pay attention to the “0th period physical education class”, which is also being implemented in Korean schools. At present, this class has not been included in the formal system of the school education system; it is implemented autonomously at the will of the principal (and very few schools are implementing the class). The class includes autonomous physical activities, in which students participate in physical activities using the morning hours (e.g., 07:30–08:30) before the commencement of the regular curriculum [1,5,13]. The “0th period physical education class” was often described as “zero hour physical education“ or “zero hour class physical activity” in papers published in Korea [8,12,13,14]. Nonetheless, the exact definition was “0th period physical education class” [15]. In foreign research, the terms “early morning physical exercises” and “early morning physical activity” were used [16,17]. This class is organized in schools in South Korea, as determined by the principal or physical education teacher [12]. Currently, in Korea, based on overseas research on the 0th period physical education class, certain elementary, middle, and high schools are voluntarily conducting zero-period physical education classes. However, most schools operate in the form that all students participate in this activity [1].

When looking at previous research on the 0th period physical education class, participation experience, perception, educational value, physical strength, and school life adaptation and satisfaction were extensively covered. However, the various benefits of the class have not been reported by the media and in overseas research. In particular, there is little research on the effect of the 0th period physical education class in improving students’ academic efficiency.

Accordingly, there is a need for continuous research on the 0th period class’s contribution to students’ academic efficiency. This positive academic interest is likely to lead to the realization of the 0th period class-related policies in the government, the Ministry of Education, and schools.

Among advanced countries, students’ participation in physical activities is strongly recommended by the government from kindergarten to high school [18,19], and studies have also reported on the benefits of morning exercise for the brain [20]. Based on this, it is necessary to find the academic meaning of the 0th period of physical education by identifying the causal relationship between the participation, perceived enjoyment, and learning attitudes of Korean teenagers, in terms of the morning exercise. In the near future, the “0th period physical education class” could be used as the basis for introducing important policy changes in the Korean school curriculum.

Therefore, the purpose of the study is to identify the possibility of “0th period physical education class” as a critical measure of encouraging morning exercise for Korean teenagers. In order to achieve the purpose of this study, this paper attempts to identify the causal relationship between participation, perceived enjoyment, and learning attitude of middle school students participating in the 0th physical education class in Korea.

### 1.1. Relationship between Perceived Enjoyment and Learning Attitude

The effects of the 0th period physical education class on participation, perceived enjoyment, and learning attitudes are of particular interest in this study. First, perceived enjoyment is a psychological concept that expresses the desire and determination to participate in sports, and is a kind of psychological satisfaction that comes from being interested in something through positive emotional reactions, such as pleasure and liking [21].

Many scholars in the fields of psychology and sports psychology have attempted theoretical approaches to enjoyment. Enjoyment is a major reason for participating in sports, regardless of the kind, participation levels, and cultural backgrounds of the students. On the contrary, if you do not have fun participating in sports, you may give up halfway [22]. This is a far-reaching concept, including internal and external—and achievement and non-achievement—dimensions, contrary to the view that enjoyment is only an indicator of internal motivation [23,24]. Scanlan et al. [25] argue that enjoyment and inner motivation are not identical concepts. That is, enjoyment in sports is a comprehensive concept. Research on perceived enjoyment in sports shows that these arguments are valid, and the feeling of fun is derived from both dimensions, as mentioned above. Therefore, external factors—such as social recognition and friendship—and internal factors—such as exercise experience and skill acquisition—can be interesting factors.

Previous literature on physical activities and perceived enjoyment is mainly related to sports, physical activity, and enjoyment in physical education classes. Wankel et al. [26] classified perceived fun in sports participation into three types: personal, social, and extrinsic factors. Furthermore, Goudas et al. [27] focused on perceived enjoyment factors accepted by students in the school’s physical education classes, and it was shown that self-deterioration influences the physical activity behavior of middle school students, which is mostly related to perceived pleasure [28]. Lewis et al. [29] noted that perceived enjoyment in physical activity is a significant factor, and the focus should be on enjoyment to promote physical activity behavior.

Second, learning attitudes refer to various attitudes related to learning that students develop during their school life. This is a psychological tendency of students to view school and learning tasks positively or negatively, which is formed throughout school life and is based on the motivation, purpose, and judgment of learning [30]. Learning attitude can be divided into learning and attitude, and learning can be considered a desirable process of change in behavior through experience and training. Acquiring knowledge and accepting the capability to make mistakes brings about a progressive change in behavior.

The definition of the learning attitude related to physical education classes is the experience of change in the learning environment through class participation. Additionally, it is the tendency of continuous and regular reaction obtained through it [31]. Fisher et al. [32] defined a self-directed learning attitude as the process of self-evaluating learning outcomes, such as diagnosing learning needs, setting goals, and securing human and material resources necessary for learning, regardless of others’ assistance.

The results of previous studies on physical education and learning attitude indicated that the scores of learning attitude were the highest in terms of intentions and actions for satisfiability of physical education in high school [33]. According to a study conducted on Korean elementary school students, early morning sports club activities were effective in increasing the learning attitude of elementary school students in South Korea [34].

The relationship between perceived fun and learning attitude is as follows. Kim [35] conducted a study on college students taking a general elective course of physical education and reported that perceived enjoyment had a positive effect on learning attitude. Moreover, the causal relationships among perceived enjoyment, learning attitude, and class satisfaction were investigated.

### 1.2. Hypotheses and Research Framework

The hypotheses of this study were established based on the relationship between participation and perceived enjoyment, perceived enjoyment and learning attitude, and participation and learning attitude. A conceptual model of the hypotheses is shown in Figure 1.

The hypotheses corresponding to each path in the conceptual model are as follows.

**Hypothesis** **1** **(H1).***Participation in the 0th period physical education class has a significant impact on perceived enjoyment*.

**Hypothesis** **2** **(H2).***Participation in the 0th period physical education class has a significant impact on learning attitude*.

**Hypothesis** **3** **(H3).***Perceived enjoyment in the 0th period physical education class has a significant impact on learning attitude*.

## 2. Methods

### 2.1. Participants

The subjects of this study were a group of middle school students in a metropolitan city in South Korea, and 300 samples were examined using a convenient sampling method. Among the survey data collected, 282 were used for the actual analysis, excluding 18 that were found to be unfaithful or missing content. The general characteristics of the subjects are presented in Table 1.

### 2.2. Measurements of Key Variables

To achieve this study’s purpose, we used structured questionnaires based on prior research and theories [31,35]. The questionnaire consisted of 32 questions, including two items on demographic characteristics, three on participation level, 12 on perceived enjoyment, and 15 on learning attitude. The questions (exciting, enjoyable, pleased, and entertaining) for measuring perceived enjoyment were adopted from previous studies [35,36]. The perceived enjoyment scale consisted of four items: exciting (three questions), enjoyable (three questions), pleased (three questions), and entertaining (three questions). The Learning Attitude Scale was measured using the questions adopted and modified from previous studies [31,37]. It consisted of four items: interest (three questions), autonomy (four questions), usefulness (four questions), and confidence (four questions). The responses to the measurement tools consisted of a five-point Likert scale (ranging from 1 = strongly disagree to 5 = strongly agree).

In this study, content validity verification of the questionnaire was conducted through consultation between two professors and three Ph.D. students majoring in sports sociology. To confirm the validity and internal consistency of the measurement tools, we conducted a confirmatory factor analysis using the maximum likelihood method and a reliability analysis using Cronbach’s α. The goodness of fit of the confirmatory factor analysis results for perceived enjoyment was χ^2^ = 213, df = 48, TLI = 0.932, CFI = 0.935, SRMR = 0.041, RMSEA = 0.047. Additionally, the goodness of fit for learning attitude was χ^2^ = 279, df = 84, TLI = 0.931, CFI = 0.935, SRMR = 0.060, RMSEA = 0.056. The outcomes are shown in Table 2.

### 2.3. Procedure and Data Analysis

To achieve the study’s objective, we visited the school and asked the physical education teacher for survey cooperation. Moreover, after explaining the purpose of the survey and how to fill it out, the survey was conducted using self-evaluation techniques. Additionally, in the case of students who had difficulty completing the questionnaire, the researcher completed it through a face-to-face survey. The data collected in this study were processed as follows. First, frequency analysis was conducted using the SPSS/PC+21.0 version statistics program for Windows. Second, we utilized the jamovi 1.2.27 program to conduct confirmatory factor analysis and reliability verification (Cronbach’s α) to verify the feasibility of the survey tool. Third, a correlation analysis was performed using SPSS/PC+21.0 version for Windows. Fourth, it was verified by conducting structural equation modeling using the Amos/PC+21.0 version statistics program for Windows. Furthermore, the probability of statistical significance was set at 0.05.

## 3. Results

### 3.1. Correlation Analysis among Variables

As shown in Table 3, the correlation between each factor determined the satisfaction of the discriminant validity between each factor for factors with single-dimensionality. Furthermore, the correlation between the relevant variables was 0.018–0.661, and there was a partially significant correlation between the variables. Since the values of all correlation numbers did not exceed 0.80, discrimination was obtained based on the criteria provided by Kline [38]. Additionally, all variables appeared smaller than 0.80—the criterion for multicollinearity between independent variables—indicating that there was no problem with multicollinearity [39].

### 3.2. Goodness of Fit of Total Effect Model

The research model of this study consisted of three variables: participation, perceived enjoyment, and learning attitude. The results of the structural equation modeling to verify the adequacy of the study’s causal model are presented in Table 4.

The goodness of fit criteria for the research model are adequate when TLI, GFI, and CFI are above 0.90, and RMSEA is below 0.080. As a result of the adequacy verification of this study, it was shown that TLI = 0.934, GFI = 0.930, CFI = 0.976, RMSEA = 0.030, which satisfied the fitted index value. Consequently, the conceptual model of this study was found to be suitable.

### 3.3. Casual Relationships between Participation, Perceived Enjoyment, and Learning Attitude in “the 0th Period Physical Education Class” of Middle Schools

The results of verifying the research hypothesis to confirm the causal relationship between perceived enjoyment, and learning attitudes in the physical education classes, are presented in Table 5.

First, participation in the 0th period of middle school had statistically significant effects on perceived enjoyment (r = 993, t = 6.092, *p* < 0.001). In other words, it was confirmed that the more students participated in sports activities during the 0th period, the higher their self-awareness of perceived enjoyment.

Second, perceived enjoyment of physical activity during the 0th period of secondary school did not have a statistically significant effect on learning attitudes (r = 0.043, t = 1.608, *p* > 0.05). Therefore, it was confirmed that the higher the perceived enjoyment of physical activities in the 0th period of school, the less the awareness of learning attitude.

Third, participation in physical activities in the 0th period of secondary school had a statistically significant effect on learning attitudes. In other words, participation in the 0th class was found to increase the perception of learning attitudes.

Finally, participation in sports activities in the 0th period of secondary school, perceived enjoyment, and learning attitude shared a causal relationship. Table 5 shows the direct, indirect, and total effects of each latent variable. Among these variables, the effect of participation in the 0th period classes on learning attitude was not only direct (*β* = 0.039), but also indirect (*β* = 0.786), indicating that the total effect was *β* = 0.825. However, perceived enjoyment did not have a significant effect on learning attitude, and statistical results confirmed that participation in 0th period physical education classes was an important variable.

## 4. Discussion

This study revolves around the hypothesis that the surprising effects of morning exercise could also be activated in the Korean education community. Based on the results of this study, we note the following.

First, the impact of participation in physical activities in the 0th period classes on perceived enjoyment was found to be significant. As such, it has been confirmed that the more middle school students participate in activities during the 0th period, the higher their self-awareness of enjoyment. More specifically, the importance of interest and fun in the activity should also be recognized as these are crucial factors in determining students’ participation in physical activities during the 0th period [40]. John and Eric [11] introduced the surprising effects of the 0th period of physical activity to the media. Through this, they emphasized the importance of activities during the 0th period, indicating that the activities are not an education, but a lifestyle. Moreover, physical activity in the morning is very effective in exercising the brain and improving students’ physical health. Moreover, it has a positive effect on emotional control by relieving stress or anxiety and reducing aggression [41]. Based on these results, this research shows that the participating students are fully satisfied with the attractiveness and benefits of physical activities in the 0th period. This is because, the internal motivation and perceived enjoyment of participating in the 0th period of physical education class have a positive impact. Portman [42] said that frequent success experiences for students can change their attitudes toward physical education and make them enjoyable. Thus, daily physical activity, such as in the 0th period, can help students build a positive personality and enjoy physical education. Students’ enthusiastic attitudes in physical education classes increases satisfaction and makes them feel more interested in the content of the class [43]. Additionally, for adolescents, successful experiences through sporting activities instill a sense of accomplishment and help build positive self-worth. Eventually, physical activity during the 0th period forms a fundamental attitude, and participation in physical activity becomes enjoyable.

Second, prior studies show that regularly participating in physical activities, including physical education classes, have a positive effect on learning immersion [44]. Additionally, the fun factor of the youth physical education classes influenced the attitude and satisfaction of the students. [45] Based on these results, applying various enjoyment factors in middle school physical education classes could increase student engagement and lead to active participation in classes. However, this study shows that perceived enjoyment among students does not have a significant impact on learning attitudes. Unlike ordinary physical education classes, perceived enjoyment is not an important factor in the 0th period physical education class; rather, participation is more important. Therefore, it is essential to plan motivational methods and classes for teenagers to encourage participation in morning exercise. These plans will not only have a positive impact on students’ learning attitudes, but also increase their capability to lead a successful school life. In fact, early morning sports club activities were effective in increasing academic achievement, learning attitude, and physical self-concept in elementary school students [34]. However, this study found a slightly different perspective. In particular, there is a need to approach elementary and middle school students with different concepts regarding their participation in the 0th period of the physical education class. This is because, in the case of elementary school students, the fun factor affects their learning attitudes, but in the case of middle school students, the enjoyment factor does not have a significant impact. Kim [46] stated that participation in physical education classes or physical activities requires research by classifying various age groups or classes. Each subject needs to be applied differently in the form of class and operation.

Third, the impact of middle school students’ participation during the 0th period on their learning attitudes was found to have a significant impact. As such, awareness of learning attitudes has been confirmed to increase depending on the degree of participation. Specifically, adolescents’ physical activity during the 0th period was found to be highly correlated with their academic performance [11], and this study showed that learning attitude was an important variable of physical activity during the 0th period. Furthermore, various studies have shown that students’ participation in sports activities has a positive relationship with academic performance [9,47,48]. Morning exercise, such as physical and mental activities in the 0th period, has a positive relationship with cognitive aspects [49]. Cardiovascular exercise through the 0th period of physical activity stimulates the production of brain cells, and its benefits include the growth of new neurons and the production of neurotropic factors (brain-derived neurotrophic factor). In other words, physical activity in the 0th period helps improve cognitive skills, such as memory, concentration, and logic [50,51,52]. In this regard, the 0th period sports activities can positively contribute to students’ physical activities in the educational environment centered on college entrance examinations in Korea. In this situation of learning, an individual’s cognitive, emotional, and behavioral response tendencies can be strengthened through physical activities, such as physical activity in the 0th period. Learning attitudes play a significant role in determining academic performance. In other words, regular exercise leads the brain to optimal conditions and influences learning [53].

If there is no interest or enjoyment in physical activity during the 0th period, it cannot be linked to a positive learning attitude. This is because physical activity in adolescence has a positive relationship with concentration and memory and contributes to increasing the speed of cognitive process [54]. Additionally, students who are actively involved in appropriate physical activities have shown holistic cognitive improvements in measuring vocabulary, memory, logic, and reaction time [52]. Therefore, regular physical activities, such as those during the 0th period, affect cognitive functions [55]. In this respect, the 0th period of physical activities will have positive effects on students’ studies, in addition to improving their physical functions.

## 5. Conclusions

This study aimed to determine the impact of middle school students’ participation in physical activities during the 0th period on perceived enjoyment and learning attitudes. Based on the analysis results, the following conclusions can be drawn:

The results of this study show that participation in the 0th period physical education class impacts perceived enjoyment and learning attitudes. However, perceived enjoyment does not affect learning attitudes. When we inferred these results, the 0th period physical education class program was not enjoyable for the students, but led them to participate consistently, which is an important factor in the learning attitude.

Thus, it seems urgent to develop practical and detailed plans to encourage middle school students to participate in the 0th period of physical education. Considering the social atmosphere of Korean parents’ one-sided educational direction (children’s college entrance exams), participation in the 0th period physical education class has a positive impact on learning attitudes, indicating the possibility that it may play an important role in changing Korean educational culture.

Future research should proceed with a study based on a variety of targets, including high school students, and a broader range of variables.

## Figures and Tables

**Figure 1 ijerph-18-07668-f001:**
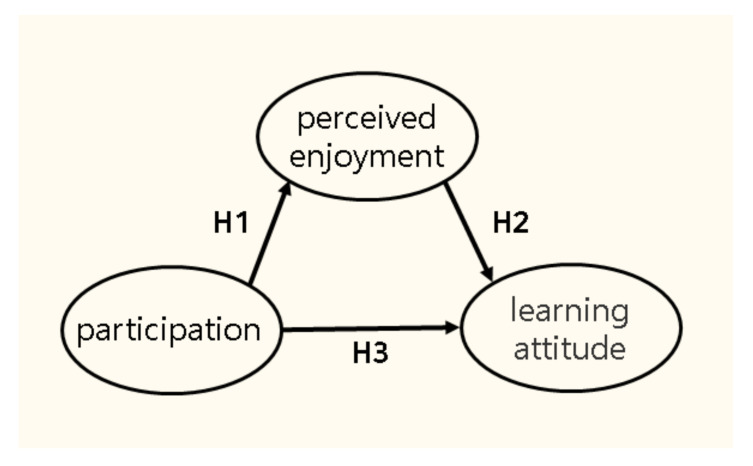
The Conceptual model.

**Table 1 ijerph-18-07668-t001:** General Feature of subject of study.

Section		Frequency	Percentage (%)
Sex	Female	161	57.1
	Male	121	42.9
School Students	First Grade	126	44.7
	Second Grade	94	33.3
	Third Grade	62	22.0

**Table 2 ijerph-18-07668-t002:** Confirmatory factor analysis and reliability of latent variables.

Factor	Variable	Measuring Variables	B	β	Standard Error	t	α
Perceived enjoyment	exciting	a01 a02 a03	1 1.040 1.029	0.971 0.954 0.958	0.031 0.030	33.00 *** 33.70 ***	0.973
	enjoyable	b01 b02 b03	1 1.077 1.111	0.918 0.899 0.942	0.055 0.047	19.30 *** 23.40 ***	0.940
	pleased	c01 c02 c03	1 1.112 1.078	0.867 0.962 0.963	0.054 0.052	20.60 *** 20.50 ***	0.948
	entertaining	d01 d02 d03	1 0.853 1.033	0.916 0.898 0.972	0.042 0.039	19.90 *** 26.20 ***	0.945
χ^2^: 213, df: 48, TLI: 0.932, CFI: 0.940, SRMR: 0.041, RMSEA: 0.047							
Learning attitude	interest	a01 a02 a03	1 0.972 0.713	0.791 0.794 0.669	0.085 0.076	11.35 *** 9.30 ***	0.793
	autonomy	b01 b02 b03 b04	1 0.097 1.075 0.996	0.699 0.716 0.472 0.438	0.122 0.202 0.205	7.97 *** 5.31 *** 4.84 ***	0.790
	usefulness	c01 c02 c03f c04	1 1.068 0.992 1.353	0.669 0.727 0.696 0.837	1 0.122 0.117 0.139	8.71 *** 8.45 *** 9.73 ***	0.825
	confidence	d01 d02 d03 d04	1 0.944 1.063 0.977	0.751 0.773 0.772 0.592	1 0.089 0.102 0.122	10.51 *** 10.42 *** 8.00 ***	0.796
χ^2^: 279, df: 84, TLI: 0.931, CFI: 0.935, SRMR: 0.060, RMSEA: 0.056							

*** *p* < 0.001.

**Table 3 ijerph-18-07668-t003:** Correlation analysis among variables.

	1	2	3	4	5	6	7	8
1	1							
2	0.627 ***	1						
3	0.594 ***	0.630 ***	1					
4	0.661 ***	0.641 ***	0.653 ***	1				
5	0.313 ***	0.282 ***	0.312 ***	0.302 ***	1			
6	0.056	0.081	0.072	0.115	0.039	1		
7	0.245 ***	0.274 ***	0.285 ***	0.253 ***	0.521 ***	0.018	1	
8	0.255 ***	0.258 ***	0.261 ***	0.274 ***	0.576 ***	0.045	0.591 ***	1

*** *p* < 0.01. 1. Exciting, 2. Enjoyable, 3. Pleased, 4. Entertaining, 5. Interest, 6. Autonomy, 7. Usefulness, 8. Confidence.

**Table 4 ijerph-18-07668-t004:** Goodness of fit of total effect model.

	χ^2^	df	CMN/df	TLI	GFI	CFI	RMSEA
**Fit index**	149.468	41	3.646	0.934	0.930	0.976	0.030
**Expropriation level**	(χ^2^/df) < 5			>0.90	>0.90	>0.90	<0.08

**Table 5 ijerph-18-07668-t005:** Path coefficient of mediated model.

	Estimates	S.E.	C.R.	Total Effects	Direct Effects	Indirect Effects	
H1	0.993	0.163	6.092 ***	0.958	0.958		Adopted
H2	0.043	0.026	1.680	0.041	0.041		Rejected
H3	0.802	0.129	6.217 ***	0.825	0.786	0.039	Adopted

*** *p* < 0.01.

## Data Availability

The data presented in this study are available on request from the corresponding authors.

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
