# Peer review of "Causal Model of Participation, Perceived Enjoyment, and Learning Attitudes in “the 0th Period Physical Education Class” of Middle Schools in South Korea"

_ijerph, 2021, doi:10.3390/ijerph18147668_

Round 1

Reviewer 1 Report

Line 76-78: In the first part of the introduction the authors describe several arguments to implement a '0th period physical education class'. In line 77 suddenly appears a new argument: "academic efficiency' Although I can endorse this, I suggest the authors to make a more firm connection with existing literature. Or to omit this argument because it will not come back later in the study

Line 90: Authors already assume that the results of the study are positive? Suppose this is not the case? Than this study does not promote it.

Line 92. The authors lack to describe the relationship between ‘academic efficiency” on one hand and participation, perceived enjoyment, and learning attitudes’ on the other hand.

Line 162: Participants: is participation in '0th period physical education class' voluntary or compulsory? When it is voluntary, this could be a major bias for he research design: only positively motivated students will join these classes.

Line 249: Authors conclude that there is a causal relationship, and that the participation in the 0th period physical education classes was an important variable (line 254). I suggest it makes a great difference whether participation i voluntary or compulsory (see previous comment)

Line 270: I suggest to add a reference for the statement ‘it has a positive effect on emotional control’

Line 352: “which serves as an important factor in changing the Korean education culture” I could not find a substantiation in the study for this firm conclusion. There may be other factors that play a much bigger role in changing an education culture.

Author Response

Dear reviewer:

First of all, I am deeply grateful for your thoughtful comments.

Line 76-78: I fully understand your opinion. Therefore, I will delete this sentence.

Line 90-92: I fully understand your opinion, too. As previously suggested, this study is not mainly aimed at the impact of 0th period physical activity on “academic efficiency”. It sought to determine the causal relationship between participation in 0th period physical education class, perceived enjoyment, and learning attitudes. Accordingly, we will modify the sentence(Lime 90-92) as follows.

“Therefore, the purpose of the study is to identify the possibility of "0th period physical education class" as a critical measure of encouraging morning exercise for Korean teenagers. In order to achieve the purpose of this study, this paper attempts to identify the causal relationship between participation, perceived enjoyment, and learning attitude of middle school students participating in the 0th physical education class in Korea."

Line 162:  Currently, in Korea, based on overseas research on 0th period physical education class, certain elementary, middle, and high schools are voluntarily conducting zero-period physical education classes. However, most schools operate in the form that all students participate in this activity. Since the explanation for this is a little lacking, I will add this in the introduction.

Line 249:  I would like to replace the answer to this with the previous explanation.

Line 270:  Thank you for your detailed assessment. we will add a reference to this sentence.

Line 352: We believe there is a problem with the expression of this sentence, so we will revise it as follows.

“ Considering the social atmosphere of Korean parents' one-sided educational direction(children's college entrance exams), participation in the 0th period physical education class has a positive impact on learning attitudes, indicating the possibility that it may playing an important role in changing Korean educational culture.”

As mentioned above, we will do our best to revise the paper.

Thank you and best Regards,

Reviewer 2 Report

The manuscript explores the perceived enjoyment and learning in the "0th-period physical education class". The topic is interesting and the research design well structured, but the manuscript could be improved with the following:  

Introduction:

  • The authors claimed that physical activity (PA) before regular classes increases brain balance and function (line 54-55), which is true, but it would also mention that any PA could improve mental health and brain function.
  • Please give more information about the Korean physical education system (how many PE they have in a week? It is mandatory or not?).
  • Please provide more information about the concepts of enjoyment because it is not clear why "exciting, enjoyable pleased entertainment" was measured in the study.

Methods:

  • Why do you need to adopt the two questionnaires? It would be necessary to clarify in the introduction or the methods.
  • Add more examples on the items in the measurement's subchapter
  • On table 2. "Ethical awareness" seems not correct

Results:

  • In table 3, the numbers are incorrect.
  • It is essential to mention that the authors are using Path Analysis instead of Structural equation modeling (SEM). SEM uses latent variables, and Path analysis assumes that all variables are measured. Please modify the name of the analysis or clarify this confusion
  • I recommend using figures to show the path coefficient instead of table 5
  • The authors argue that enjoyment has external and internal factors. However, this concept did not appear in the empirical study. It would be interesting to see the relationship between the different subscales. Why did you choose to analyze the scale as a whole? It would be also interesting to see what the relations between the subscales are.

Discussion and conclusions

  • I believe the lines between 273 and 285 should be rephrased. The authors explain the benefits of the 0th period PE classes. However, these benefits could be appearing in regular PE classes as well.

Author Response

Dear reviewer:

First of all, I am deeply grateful for your thoughtful comments. The responses to your review are as follows, and we have tried to revise it as much as possible.

Introduction:

Currently, in Korea, based on overseas research on 0th period physical education class, certain elementary, middle, and high schools are voluntarily conducting zero-period physical education classes. However, most schools operate in the form that all students participate in this activity. Since the explanation for this is a little lacking, I will add this in the introduction.

Method:

There was a mistake in the description of table 2. A correction was made for this. Thank you very much.

Result:

There was a mistake in the description of table 3. A correction was made for this. Thank you very much, too.

In addition, The hypothesis established by this author is that the total, direct, and indirect effects of variables that have causal relationships between the exogenous and endogenous variables of the unobserved variable were seen. It is also presented based on the atandized estimates of the direct effects. Therefore, it was analyzed based on a structural equation model rather than path analysis based on regression analysis.

Discussion and conclusions:

I fully understand what you are referring to. Also, I bet that the sentences may cause misunderstanding. Therefore, these sentences have been revised as follows as a whole.

“First, the impact of participation in physical activities in the 0th period classes on perceived enjoyment was found to be significant. As such, it has been confirmed that the more middle school students participate in activities during the 0th period, the higher their self-awareness of enjoyment. More specifically, the importance of interest and fun in the activity should also be recognized as these are crucial factors in determining students' participation in physical activities during the 0th period [40]. John and Eric [11] introduced the surprising effects of the 0th period of physical activity to the media. Through this, they emphasized the importance of activities during the 0th period, indicating that the activities are not an education but a lifestyle. Moreover, physical activity in the morning is very effective in exercising the brain and improving students' physical health. Moreover, it has a positive effect on emotional control by relieving stress or anxiety and reducing aggression [49]. Based on these results, this research shows that the participating students are fully satisfied with the attractiveness and benefits of physical activities in the 0th period. This is because, the internal motivation and perceived enjoyment of participating in the 0th period of physical education class have a positive impact. Portman [41] said that frequent success experiences for students can change their attitudes toward physical education and make them enjoyable. Thus, daily physical activity, such as in the 0th period, can help students build a positive personality and enjoy physical education. Students' enthusiastic attitude in physical education classes increases satisfaction and makes them feel more interested in the contents of the class [43]. Additionally, for adolescents, successful experiences through sports activities instill a sense of accomplishment and help build a positive self-worth. Eventually, physical activity during the 0th period forms a fundamental attitude, and participation in physical activity becomes enjoyable.”

As mentioned above, we will do our best to revise the paper.

Thank you and best Regards,
